# The Importance of Clinical Examination under General Anesthesia: Improving Parametrial Assessment in Cervical Cancer Patients

**DOI:** 10.3390/cancers13122961

**Published:** 2021-06-13

**Authors:** Paulina Sodeikat, Massimiliano Lia, Mireille Martin, Lars-Christian Horn, Michael Höckel, Bahriye Aktas, Benjamin Wolf

**Affiliations:** 1Department of Gynecology, Leipzig University Medical Center, D-04103 Leipzig, Germany; paulina.sodeikat@web.de (P.S.); massimiliano.lia@medizin.uni-leipzig.de (M.L.); michael.hoeckel@medizin.uni-leipzig.de (M.H.); bahriye.aktas@medizin.uni-leipzig.de (B.A.); 2Department of Diagnostic and Interventional Radiology, Leipzig University Medical Center, D-04103 Leipzig, Germany; mireille.martin@medizin.uni-leipzig.de; 3Division of Gynecologic, Breast, and Perinatal Pathology, Leipzig University Medical Center, D-04103 Leipzig, Germany; lars-christian.horn@medizin.uni-leipzig.de

**Keywords:** cervical cancer, magnetic resonance imaging, examination under anesthesia, accuracy, sensitivity, specificity, predictive value

## Abstract

**Simple Summary:**

In most cases, the treatment strategy (radiation or surgery) in cervical cancer patients depends on whether the parametrium shows tumor involvement. Traditionally, clinical pelvic examination under general anesthesia (EUA) has been used to determine whether tumor spread into the parametrium is present. During the recent decade, however, magnetic resonance imaging (MRI) has been increasingly used to determine whether parametrial tumor extension is present, and several studies have indicated that MRI might be superior to EUA. In this study, we demonstrate that EUA still plays an important role in pre-therapeutic evaluation of cervical cancer patients, and that display of MR images in the operating room (augmented EUA) achieves superior results in predicting parametrial tumor spread when comparted to MRI alone, especially in larger tumors. Best predictive results were observed in cases when radiologists and gynecological oncologists agreed on parametrial status, highlighting the importance of interdisciplinary patient assessment.

**Abstract:**

Background: Parametrial tumor involvement is an important prognostic factor in cervical cancer and is used to guide management. Here, we investigate the diagnostic value of clinical examination under general anesthesia (EUA) and magnetic resonance imaging (MRI) in determining parametrial tumor spread. Methods: Post-operative pathological findings of 400 patients with primary cervical cancer were compared to the respective MRI data and the results from EUA. The gynecological oncologist had access to the MR images during clinical assessment (augmented EUA, aEUA). Results: Pathologically proven parametrial tumor invasion was present in 165 (41%) patients. aEUA exhibited a higher accuracy than MRI alone (83% vs. 76%; McNemar’s odds ratio [OR] = 2.0, 95%CI 1.25–3.27, *p* = 0.003). Although accuracy was not affected by tumor size in aEUA, MRI was associated with a lower accuracy in tumors ≥2.5 cm (OR for a correct diagnosis compared to smaller tumors 0.22, *p* < 0.001). There was also a decrease in specificity when evaluating parametrial invasion by MRI in tumors ≥2.5 cm in diameter (*p* < 0.0001) compared to smaller tumors (< 2.5 cm). Body mass index had no influence on performance of either method. Conclusions: aEUA has the potential to increase the diagnostic accuracy of MRI in determining parametrial tumor involvement in cervical cancer patients.

## 1. Introduction

In 2019, the Fédération Internationale de Gynécologie et d’Obstétrique (FIGO) released a revised cervical cancer staging system which calls for the incorporation of imaging modalities for the first time [1]. Studies suggest that evaluation of cervical cancer using magnetic resonance imaging (MRI) is superior to evaluation by clinical examination in earlier cancer stages (IB1-IIA2) [2,3,4,5,6]. However, the roles of MRI and clinical examination in advanced stages, characterized by tumor extension beyond the uterine cervix, remain unclear. Especially tumor spread into the laterally abutting supporting tissues, the so called parametria (stage IIB), is of importance as it is usually considered to be a contraindication for surgical treatment. In a systematic literature review of studies published between 2012 and 2018, Woo et al. [5] found that sensitivity for detection of parametrial infiltration (PMI) by MRI was 76% and specificity was 94%. These numbers are concordant with two meta analyses which included studies conducted within the past 30 years [4,7]. Of note, only four out of the 14 studies included in these analyses involved patients with cancer staged IIB or higher. In contrast, sensitivity for the detection of parametrial infiltration by clinical examination varies between 29% and 66% [4,5,8,9,10,11]. In the reported data, specificities for detecting parametrial invasion by clinical examination versus magnetic resonance imaging range from 81% to 99% for clinical examination and from 63% to 99% for MRI [8,9,12,13,14]

Most studies investigating parametrial tumor involvement are limited by the circumstance that comprehensive post-operative histology for locally advanced cases is not available as patients with presumed parametrial infiltration undergo primary chemo-radiotherapy at most institutions, in accordance with current national and international guidelines [15,16].

Since parametrial invasion (PMI) is an established factor governing treatment decisions [15] and prognosis [17,18], more information regarding the value of MR imaging and clinical examination to detect parametrial cancer spread is needed. During the past two decades, we conducted the Leipzig School Mesometrial Resection (MMR) study, which allowed us to treat patients with locally advanced cervical cancer surgically [19]. This enabled us to compare clinical staging and MRI using post-operative histology as a reference for cervical cancer up to stage IVA. The aim of this retrospective analysis was to compare the sensitivity, specificity, and predictive values for detection of PMI between MRI and clinical examination under anesthesia (EUA) in cervical cancer stages IB1-IVA, and to determine if factors, such as tumor size or body mass index (BMI), influence test performance of either modality.

## 2. Materials and Methods

### 2.1. Patient Selection 

For this retrospective analysis, we identified patients with cervical cancer staged IB to IVA (according to the 2009 FIGO staging criteria [20]) in our study database, who had undergone primary surgical treatment at the University Hospital Leipzig between 10/2000 and 07/2017. All patients were participants of the prospective monocentric observational MMR study at our institution which was initiated in September 1999 to evaluate a novel surgical strategy for the treatment of cervical cancer based on the theory of ontogenetic cancer fields. The study was approved by our institutional ethics review board and was registered retrospectively at the German Clinical Trials Registry (DRKS00015171). A detailed description of the trial has been published [19] and is available at https://www.drks.de/drks_web/navigate.do?navigationId=trial.HTML&TRIAL_ID=DRKS00015171 (accessed on 12 June 2021). All patients provided informed consent to participate in this study which included the permission to use data for further analysis. Besides the exclusion criteria specified in the MMR study protocol, such as previous major pelvic surgery and the presence of severe systemic disease prohibiting surgery (American Society of Anesthesiologists [ASA] score ≥3), for this current analysis we also excluded women who had undergone neoadjuvant treatment with chemotherapy and patients for whom no MRI-reports were available for review. Therefore, all patients included in this study had undergone preoperative MRI and had been submitted to EUA. In addition, patients had received cystoscopy and rectoscopy when deemed appropriate by the examiner. Importantly, MR imaging was performed before EUA and the images were displayed in the operating room (augmented EUA, aEUA). The radiology reports (i.e., the written interpretation of the MR images by a board-certified radiologist) were not available during clinical assessment. Throughout this text, aEUA refers to clinical assessment under general anesthesia with synchronous display of the MR images. Of importance, aEUA was performed as a separate procedure before definitive surgical treatment and aEUA does not refer to intra-operative parametrial assessment during radical hysterectomy. The diagnostic information relevant for this current analysis (i.e., data from MRI and aEUA) were gathered retrospectively from our written records. We compared MRI and clinical findings specifically focusing on parametrial involvement. The results were then compared with the pathology reports, which were set as the reference standard. Factors which might influence the accuracy of parametrial assessment were analyzed, such as a patient’s body mass index (BMI) and tumor size. In addition, we investigated whether the sensitivity and specificity of parametrial assessment by one examination method could be improved by case stratification according to the test results of the other examination method. For example, we investigated whether detection of parametrial infiltration by MRI was more reliable in the subgroup of patients with parametrial involvement found on aEUA.

### 2.2. Statistical Analysis

Details on statistical analysis can be found in the Appendix A. In brief, continuous data are presented in medians and inter-quartile ranges (IQR) while categorical data are given as percentages. Confidence intervals (CI), when applicable, are given for the 95% range. The sensitivity and specificity for the detection of parametrial tumor invasion of each method of examination, as well as negative predictive value (NPV) and positive predictive value (PPV) were calculated. Accuracy of both MRI and aEUA was compared using the exact McNemar’s test. To determine whether the differences in test performance between MRI and aEUA were significant, we applied McNemar’s test for paired data to calculate χ^2^ and *p*-values. To determine the statistical significance of the differences in PPV and NPV for the paired samples, we used the relative predictive value function as proposed by Moskowitz and Pepe. Fisher’s exact test was used to calculate statistical significance of differences in sensitivity, specificity, PPV, and NPV between different groups of patients assessed with the same diagnostic method (i.e., unpaired samples, e.g., patients with tumors < 2.5 cm compared to patients with tumors ≥2.5 cm assessed for parametrial infiltration by MRI). By convention, the differences in the outcome of both methods were considered statistically significant with a *p*-value of 0.05 or less. To ascertain the relevance of potentially influencing factors (BMI, tumor size, parametrial status as assessed by the other examination method) on the accuracy of aEUA and MRI, we built univariable logistic regression models. In addition, we computed a multivariable regression model including all parameters from the univariable regression. 

## 3. Results

### 3.1. Patient characteristics

During the study period from 10/2000–05/2017, 551 patients were treated surgically for primary cervical cancer, of which 400 met the inclusion criteria of our study (see Figure 1). A summary of the clinicopathological characteristics of all 400 patients is given in Table 1 (pathology data is from analysis of the post-operative specimen). 

The median time interval between acquisition of MR images and aEUA was 1 day (IQR 0–1, range 0–49). In total, 349 patients (87.3%) had MR images taken on the day of aEUA or the day before. In 5 cases (1.3%), the time elapsed between MRI and aEUA was more than 10 days. In all these cases, MRI and aEUA findings regarding parametrial assessment did not differ.

Parametrial tumor invasion (stage ≥ IIB) was detected in 39.3% clinically, in 39.3% radiologically, and in 41.3% pathologically. 

### 3.2. Assessment for Parametrial Involvement by aEUA Versus MRI

The distribution of positive and negative test results within our study population is depicted in Figure 2. aEUA exhibited a higher accuracy (83%) as compared to MRI (76%). This difference was statistically significant (McNemar’s OR = 2.0, 95%CI 1.25–3.27, *p* = 0.003). Further test results for sensitivity, specificity, and negative and positive predictive values are provided in Table 2. In summary, aEUA showed statistically significant higher sensitivity, specificity, PPV, and NPV regarding tumor involvement of the parametrium compared to MRI alone.

### 3.3. Tumor Size as an Influencing Factor on Test Performance

For this analysis we used tumor sizes as determined by MRI and we used 2.5 cm maximal tumor diameter as a cut-off value. The cut-off value was chosen based on findings from Woo et al. [21]. Exact tumor size as determined by MRI was available and explicitly stated in the radiology reports of 297 (74.3%) patients. Of these cases, 77 (25.9%) had tumors <2.5 cm in diameter, and in 220 (74.1%) tumors measured ≥2.5 cm. In the subgroup of patients with tumors <2.5 cm, pathologically proven parametrial infiltration was present in 10 (30%) of patients. In the subgroup comprising patients with tumors ≥2.5, the prevalence of parametrial infiltration was 61.4%. In summary, specificity and NPV of parametrial assessment by MRI were significantly better in smaller tumors. In aEUA, PPV was better in larger and NPV in smaller tumors (Table 3 and Table 4). Although accuracy was not affected by tumor size in aEUA, MRI was associated with a significant drop in accuracy in tumors ≥2.5 cm (univariable logistic regression, OR for a correct diagnosis compared to smaller tumors 0.22, *p* < 0.001). This association remained significant in a multivariable regression model (Table 5).

### 3.4. The Influence of BMI on Testing for Parametrial Invasion

To evaluate the influence of BMI on sensitivity and specificity for the assessment of parametrial involvement, we compared results of 337 (84.3%) patients with a BMI < 30 kg/m^2^ to those of 55 (13.8%) patients with a BMI ≥ 30 kg/m^2^. In 8 cases (2%) the patient’s BMI was unknown. Median tumor size in women presenting with a BMI ≥ 30 kg/m^2^ was 4 cm. Women with a BMI less than 30 kg/m^2^ had tumors with a median size of 3.7 cm. Diagnostic performance measures are presented in Table 3 and Table 4. Taken together, BMI had no influence on test performance in either clinical or MRI examination. The same was evident for accuracy assessment by univariable and multivariable logistic regression modeling (Table 5).

### 3.5. Incremental Value of Performing Both, Clinical and Radiological Assessment

We found that sensitivity and PPV of MRI increased from 39.5% and 35.7%, respectively, in cases without clinical evidence of parametrial involvement (*n* = 243, 60.8%) to 77.2% and 85.2% in tumors with clinical evidence of parametrial infiltration (*n* = 157, 39.2%, *p* < 0.0001 for both, sensitivity and PPV). In contrast, specificity and NPV were significantly lower in cases with parametrial infiltration suspected on aEUA (Table 3). Likewise, we analyzed whether aEUA results were better in patients with MRI evidence of parametrial infiltration. Sensitivity and PPV of aEUA was higher in patients with tumors radiologically staged < cT2b, and, accordingly, lower in patients with tumors staged cT2b and higher (Table 4). Specificity and NPV declined from 93.2% and 88.7% for stages < cT2b to 61.4% and 64.3% for stages ≥ cT2b (*p* < 0.0001 for both measures).

### 3.6. Accuracy and Net-Sensitivity and Specificity with Combined Examination

Highest diagnostic accuracy measures were observed when there was agreement regarding parametrial involvement among the two assessment methods. In 315 (78.8%) congruent cases, parametrial involvement was recognized with a sensitivity of 81.7% and a specificity of 91.3%. Net sensitivity, i.e., the probability of recognizing true parametrial involvement with a positive result in either MRI, aEUA, or both methods, was 86.1%. Net specificity, i.e., the probability of correctly identifying the absence of parametrial involvement when both MRI and clinical assessment were negative, was 92.8%. Logistic regression models demonstrated that accuracy was significantly higher in the subgroup of patients with concordant results (OR of achieving correct results of 7.5 and 6.0 on univariable and multivariable regression modelling, respectively, *p* < 0.0001 in both cases, Table 5).

### 3.7. Change of Diagnostic Test Performance over Time

As the study period spanned almost two decades, we assessed whether diagnostic performance of either method changed over time. We arbitrarily defined three time periods, each spanning 5–6 years, and determined whether accuracy changed over time. There was not a significant change in the accuracy of eather method between the time periods (*p* = 0.06 for aEUA and *p* = 0.34 for MRI, see Figure 3). 

## 4. Discussion

This current study was conducted to investigate the accuracy of parametrial assessment in patients with cervical cancer using MRI and clinical examination under general anesthesia. We introduce aEUA, i.e., clinical examination under general anesthesia augmented by display of MR images in the operating room, as a novel concept to improved cervical cancer staging. We found that aEUA results in higher accuracy, sensitivity, specificity, as well as better negative and positive predictive values than MRI alone. 

Our findings need to compared to those of other investigators who generally report higher accuracy and better performance of MRI [2,4,8,10,11]. Important limitations to all these studies are, however, that they included only small numbers of patients with tumors staged IIB or higher, and in most cases with IIB disease, no pathological assessment of the parametrium was performed as these patients were generally treated by primary radiotherapy. In one former meta-analysis, pooled sensitivity of PMI assessment in clinical examination was 40% [4]. However, only four studies with post-operative histopathological assessment from the 1980s, comprising 81 cases in total, included patients with tumors staged IIB or higher [4]. The author’s conclusion, that sensitivity of clinical examination was probably underestimated due to a low prevalence of PMI and advanced stages, is undermined by results from our investigation, as we found that sensitivity for clinical examination was 77%. It should be noted, however, that the comparison of our data with that of the aforementioned study is limited by the circumstance that we did not strictly compare clinical examination and MRI, but rather aEUA and MRI alone. The improved sensitivity observed in our study might, therefore, also be a consequence of the different assessment methods.

On the other hand, in our study sensitivity of MRI (68%) was lower than reported elsewhere, where it ranges up to 100%. Regarding specificity, the difference between aEUA and MRI was less pronounced, though still statistically significant (81.4% [95% CI: 76.3–86.3] vs. 87.2% [95% CI 83.0–91.5], *p* = 0.027). Previous studies found comparatively good sensitivity using both examination methods [4,5,10,14,22]. 

Both specificity and NPV of PMI assessment by MRI were significantly better in smaller tumors (<2.5 cm). In MRI, parametrial invasion is suspected when disruption of the normal hypointense signal of the cervical stromal ring is present [23], and in tumors with greater size, it is more difficult to distinguish peritumoral edema from actual tumor invasion, leading to an overestimation of tumor stage [24]. Therefore, the higher specificity and NPV in smaller tumors which we observed in our study can be well explained. Because prevalence directly affects predictive values, PPV and NPV are expected to vary with tumor size. In addition, sensitivity and specificity have also been shown to be affected by the prevalence of a given feature (parametrial infiltration) through other mechanisms, though clearly less so than predictive values [25]. Indeed, many studies show a positive correlation between tumor size and presence of PMI [26,27,28,29], a finding which our data confirms. An important result of our study is, however, that aEUA in contrast to MR imaging alone seems to be less affected by tumor size, as shown by comparison of sensitivity, specificity, and logarithmic regression assessing accuracy (Table 3, Table 4 and Table 5). This result is best explained by the better performance of clinical palpation in larger tumors, augmented by display of the MR images in the operating room enabling the gynecological oncologist to directly correlate tactile and visual information. Differences in NPV and PPV are plausible considering the higher prevalence of PMI in larger tumors. 

There are few studies that investigate the influence of obesity on staging modalities. Although Uppot et al. suggest a lower image quality in obese women [30], there was neither a significant difference in sensitivity and specificity assessing PMI clinically in women with body mass index ≥30 kg/m^2^ compared to those with BMI < 30 kg/m^2^ nor when evaluating PMI with MRI. In fact, there was rather a non-significant increase in sensitivity of PMI detection in both examination methods when patients presented with a BMI over 30 kg/m^2^. This observation, however, is likely attributable to a larger median tumor size in this subgroup of patients (4 cm) which is concordant with our observation that sensitivity is higher in larger tumors. 

Best results in sensitivity and specificity for PMI assessment were observed in the subgroup of cases where gynecologists and radiologists agreed on parametrial tumor status, which is consistent with observations made by other investigators [8,12,31]. This was also apparent in our univariable and multivariable regression models which demonstrate that accuracy is significantly higher in patients with concordant results. Our results emphasize, therefore, the importance of interdisciplinary case assessment when staging cervical cancer patients.

One aspect of our study regarding the statistical analyses deserves special consideration: We used McNemar’s test to calculate OR’s to compare accuracy, as well as sensitivity, specificity, and predictive values of MRI and aEUA. In addition, relative predictive values were calculated. For example, considering the accuracy of MRI and aEUA (76% vs. 83%), McNemar’s OR was found to be 2.0. Counterintuitively, this does not indicate that aEUA results are twice as often correct as MRI results. Rather, because McNemar’s test only considers cases with divergent test results, it means that in cases when MRI and aEUA results are conflicting, aEUA findings will be correct twice as often as MRI findings.

Some limitations of our study should be considered when interpreting these results. First, this study was not a strict comparison of EUA and MR-imaging, as the gynecologic oncologist assessing the patient had access to the MR-images (termed aEUA). This enables the examiner to correlate palpatory findings with pelvic images, thus refining the diagnostic process. At the same time, the radiologists were not routinely informed about clinical findings. Second, our study period spans two decades, and significant progress has been made regarding MRI technology and image analysis. Therefore, our findings might not entirely reflect MRI quality achievable today. Indeed, new algorithms for MR image acquisition and analysis have been established at our institution during the past two years and their effect on diagnostic accuracy will be evaluated prospectively in the near future. However, MRI accuracy did not change significantly over time (Figure 3). Third, it was not a multicentric, prospective study which limits the overall generalizability of our results. Yet, our findings stress the importance of adequate training in attentive, bimanual, rectovaginal palpation of gynecological patients. Last, our study does not provide insight why assessment of larger tumors by MRI is less accurate. It might be related to peritumoral inflammation, but this has to be investigated in future studies.

Significant strengths of our study include that all women underwent surgery and 41% had pathologically proven parametrial involvement. Furthermore, the number of cases is large compared with most other studies. Moreover, selection bias is minimized because staging did not affect therapy choice.

As this is a retrospective study, prospective evaluation of the aEUA concept is needed. We are currently planning such an investigation at our institution. Further study regarding the inferior accuracy of MRI in larger tumors as compared to smaller tumors is also needed. It would be especially interesting to evaluate whether peritumoral inflammation contributes to reduced specificity in larger tumors. 

As most women with locally advanced cervical cancer are treated with primary chemoradiotherapy, the majority of patients with suspected parametrial tumor infiltration will not receive surgery irrespective whether parametrial tumor extension is suspected on MRI or aEUA. However, our findings should be kept in mind when interpreting MR images from cervical cancer patients, especially with larger tumors. Irrespective of the treatment pursued, optimal pathoanatomical diagnosis of the disease is mandatory.

## 5. Conclusions

We demonstrate that clinical evaluation of the parametrium under general anesthesia with display of MR images in the operating room, performed by a gynecologic oncologist, is more accurate than MR imaging alone in detecting parametrial tumor involvement by cervical cancer. Display of MR images in the operating room for assessment by the examining gynecologic oncologist should, therefore, be added to standard examination under anesthesia. In addition, our data shows that MRI assessment of smaller tumors (< 2.5 cm) is more reliable compared to larger tumors. BMI does not affect diagnostic accuracy of either clinical or MRI examination. Future prospective studies need to confirm these findings.

## Figures and Tables

**Figure 1 cancers-13-02961-f001:**
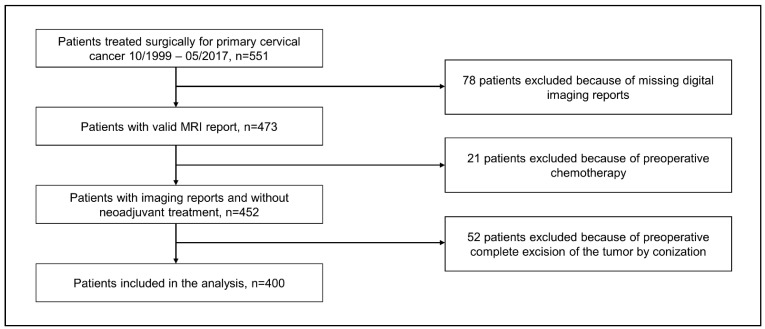
Flowchart depicting the patient selection process.

**Figure 2 cancers-13-02961-f002:**
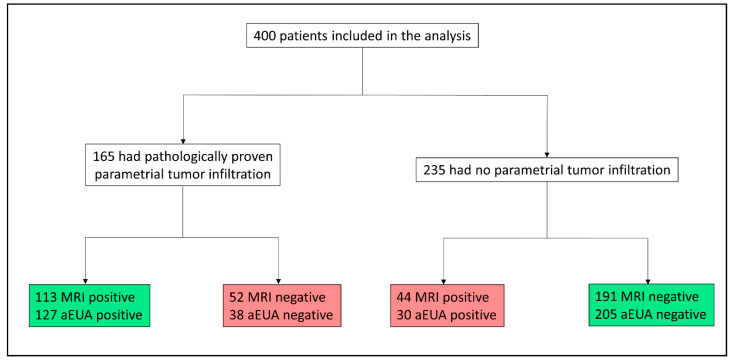
Population tree demonstrating the diagnostic results (MRI and aEUA) within our study population. All accuracy calculations are based on these numbers. aEUA: clinical examination under general anesthesia with display of MR images in the operating room. MRI: magnetic resonance imaging.

**Figure 3 cancers-13-02961-f003:**
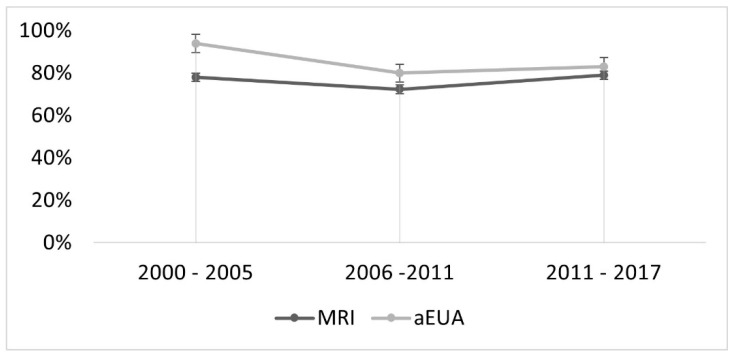
Line graph indicating change in accuracy of MRI and aEUA over time. There was no significant change for both methods (*p* = 0.06 for aEUA and *p* = 0.34 for MRI).

**Table 1 cancers-13-02961-t001:** Patient and tumor characteristics.

Parameter		*n*	%
Number of patients		400	100
Median age, in years (IQR)		46	37–55.5
Median BMI, in kg/m^2^ (IQR)		23	21–27
Histological type	Squamous cell carcinoma	302	75.5
	Adenocarcinoma	75	18.75
	Adenosquamous carcinoma	19	4.75
	Neuroendocrine carcinoma	2	0.5
	Other	2	0.5
FIGO stage	IB1	174	43.5
	IB2	36	9
	IIA1	16	4
	IIA2	17	4.25
	IIB	136	34
	IIIA	2	0.5
	IIIB	12	3
	IVA	7	1.75
Median tumor size, cm (IQR)		3.7	2.7–4.9
pT-stage	pT1a	1	0.25
	pT1b1	159	39.75
	pT1b2	56	14
	pT2a1	12	3
	pT2a2	7	1.75
	pT2b	155	38.75
	pT3b	2	0.5
	pT4	8	2
MRI stage (cT)	cT0	51	12.75
	cT1b1	123	30.75
	cT1b2	17	4.25
	cT2a	52	13
	cT2b	138	34.5
	cT3a	2	0.5
	cT3b	1	0.25
	cT4	16	4
Grading	G1	58	14.5
	G2	187	46.75
	G3	150	37.5
	G4	1	0.25
	Unknown	4	1
Lymphovascular involvement	Yes	276	69
	No	121	30.25
	Unknown	3	0.75
Blood vessel involvement	Yes	62	15.5
	No	336	84
	Unknown	2	0.5
Pelvic lymph node metastasis	pN0	247	61.75
	pN1	153	38.25
Paraaortic lymph node metastasis	pM0	364	91
	pM1	36	9

BMI: body mass index. FIGO: Fédération Internationale de Gynécologie et d’Obstétrique. IQR: Interquartile range. MRI: magnetic resonance imaging.

**Table 2 cancers-13-02961-t002:** Diagnostic performance metrics of MRI and aEUA for pathologically confirmed parametrial infiltration.

Measure	MRI			aEUA			OR			
	Estimate	Lower 95% CI	Upper 95% CI	Estimate	Lower 95%-CI	Upper 95% CI	Estimate	Lower 95%-CI	Upper 95%-CI	*p*-Value
Sensitivity	68.5	61.4	75.6	77.0	70.5	83.4	1.93 *	1.01	3.88	0.048 *
Specificity	81.4	76.3	86.3	87.2	83.0	91.5	2.08 *	1.04	4.38	0.038 *
PPV	72.0	64.9	79.0	80.9	74.7	87.0	1.12 **	1.03	1.22	0.007 **
NPV	78.6	73.4	83.8	84.4	79.8	88.9	1.07 **	1.02	1.13	0.012 **
Accuracy	76.0	71.5	80.1	83.0	79.0	86.6	2.0	1.25	3.27	0.003 *

* McNemar’s OR, ** Relative predictive value, MRI: magnetic resonance imaging. aEUA: examination under anesthesia with display of MR images (augmented EUA). CI: confidence interval. NPV: negative predictive value. OR: odds ratio. PPV: positive predictive value.

**Table 3 cancers-13-02961-t003:** Factors influencing the diagnostic performance of MRI in detecting pathologically confirmed parametrial tumor involvement.

Tumor Size (MRI)	<2.5 cm (*n* = 77)			≥2.5 cm (*n* = 220)			OR			*p*-Value
	Estimate	Lower 95%CI	Upper 95%CI	Estimate	Lower 95%CI	Upper 95%CI	Estimate	Lower 95%CI	Upper 95%CI	
Sensitivity	60	29.64	90.36	70.37	62.67	78.07	1.59	0.31	7.14	0.4917
Specificity	92.54	86.24	98.83	58.82	48.36	69.29	0.12	0.03	0.33	**<0.0001**
PPV	54.55	25.12	83.97	73.08	65.45	80.7	2.22	0.5	10.0	0.29
NPV	93.94	88.18	99.7	55.56	45.29	65.82	0.08	0.02	0.25	**<0.0001**
Accuracy	89.47	80.31	95.34	65.91	59.24	72.15	0.26	0.11	0.55	**<0.001**
**BMI**	**<30 kg/m^2^ (*n* = 337)**			**≥30 kg/m^2^ (*n* = 55)**			**OR**			***p*-Value**
	**Estimate**	**Lower 95%CI**	**Upper 95%CI**	**Estimate**	**Lower 95%CI**	**Upper 95%CI**	**Estimate**	**Lower 95%CI**	**Upper 95%CI**	
Sensitivity	66.9	59.16	74.64	80.95	64.16	97.75	2.08	0.63	9.09	0.2201
Specificity	81.01	75.52	86.53	79.41	65.82	93.0	0.90	0.35	2.65	0.8154
PPV	71.97	64.31	79.63	70.83	52.65	89.02	0.94	0.34	2.96	1.0
NPV	77.07	71.32	82.83	87.1	75.3	99.0	2	0.65	8.27	0.25
Accuracy	75.07	70.10	79.60	80.0	67.03	89.57	1.33	0.64	2.99	0.4997
**Clinical Parametrial Status**	**Negative (*n* = 243)**			**Positive (*n* = 157)**			**OR**			***p*-Vlaue**
	**Estimate**	**Lower 95%CI**	**Upper 95%CI**	**Estimate**	**Lower 95%CI**	**Upper 95%CI**	**Estimate**	**Lower 95%CI**	**Upper 95%CI**	
Sensitivity	39.47	23.93	55.01	77.17	69.86	84.47	5.0	2.22	12.5	**<0.0001**
Specificity	86.83	82.2	91.46	43.33	25.60	61.07	0.12	0.05	0.29	**<0.0001**
PPV	35.71	21.22	50.21	85.22	78.73	91.70	12.5	5.26	33.3	**<0.0001**
NPV	88.56	84.16	92.96	30.95	16.97	44.93	0.06	0.02	0.14	**<0.0001**
Accuracy	79.42	73.79	84.33	70.70	62.92	77.68	0.63	0.38	1.03	0.054

BMI: body mass index. CI: confidence interval. MRI: magnetic resonance imaging. NPV: negative predictive value. OR: odds ratio. PPV: positive predictive value. Bold: highlight significant values.

**Table 4 cancers-13-02961-t004:** Factors influencing diagnostic performance of aEUA in detecting pathologically confirmed parametrial tumor infiltration.

Tumor Size (MRI)	<2.5 cm (*n* = 77)			≥2.5 cm (*n* = 220)			OR			*p*-Value
	Estimate	Lower 95%CI	Upper 95%CI	Estimate	Lower 95%CI	Upper 95%CI	Estimate	Lower 95%CI	Upper 95%CI	
Sensitivity	70.0	41.6	98.4	77.78	70.76	84.79	0.67	0.14	4.25	0.6954
Specificity	89.56	82.23	96.88	80.0	71.5	88.5	0.47	0.15	1.29	0.1223
PPV	50.0	23.81	76.19	86.07	79.92	92.21	0.16	0.04	0.63	**0.003**
NPV	95.24	90.0	100	69.39	60.26	78.51	0.16	0.02	0.4	**<0.0001**
Accuracy	87.01	77.41	93.59	78.64	72.62	83.86	0.55	0.23	1.19	0.1307
**BMI**	**<30 kg/m^2^ (*n* = 243)**			**≥30 kg/m^2^ (*n* = 157)**			**OR**			***p*-Value**
	**Estimate**	**Lower 95%CI**	**Upper 95%CI**	**Estimate**	**Lower 95%CI**	**Upper 95%CI**	**Estimate**	**Lower 95%CI**	**Upper 95%CI**	
Sensitivity	75.35	68.26	82.44	85.71	70.75	100	0.51	0.09	1.91	0.4105
Specificity	86.67	81.9	91.44	88.24	77.41	99.07	1.15	0.36	4.87	1.0
PPV	80.45	73.71	87.19	81.82	65.7	97.94	0.91	0.21	3.11	1.0
NPV	82.84	77.67	88.02	90.91	81.1	100	2.1	0.59	11.16	0.31
Accuracy	81.90	77.37	85.86	87.27	75.52	94.73	1.51	0.64	4.17	0.4422
**MRI Parametrial Status**	**Negative (*n* = 243)**			**Positive (*n* = 157)**			**OR**			***p*-Vlaue**
	**Estimate**	**Lower 95%CI**	**Upper 95%CI**	**Estimate**	**Lower 95%CI**	**Upper 95%CI**	**Estimate**	**Lower 95%CI**	**Upper 95%CI**	
Sensitivity	55.77	42.27	69.27	86.73	80.47	92.98	0.2	0.08	0.45	**<0.0001**
Specificity	93.19	89.62	96.77	61.36	46.98	75.75	0.12	0.05	0.29	**<0.0001**
PPV	69.05	55.07	83.03	85.22	78.73	91.70	0.39	0.16	0.98	**0.04**
NPV	88.56	84.16	92.96	64.29	49.79	78.78	0.23	0.1	0.55	**0.0003**
Accuracy	85.19	80.08	89.40	79.62	72.46	85.62	0.68	0.39	1.20	0.1729

BMI: body mass index. AEUA: Examination under anesthesia with display of MR images. CI: confidence interval. MRI: magnetic resonance imaging. NPV: negative predictive value. OR: odds ratio. PPV: positive predictive value. Bold: highlight significant values.

**Table 5 cancers-13-02961-t005:** Factors influencing accuracy of parametrial assessment by MRI and aEUA: regression modeling.

Univariable Regression Modeling						
**MRI**						
Parameter	*n*	Estimate	Standard error	z-value	OR	*p*
Size (<2.5 cm vs. ≥2.5 cm)	296	−1.501	0.3997	−3.755	0.22	**0.000173**
BMI (<30 kg/m^2^ vs. ≥30 kg/m^2^)	392	0.2717	0.3601	0.755	1.31	0.45
Parametrial status (clinical exam, negative vs. positive)	400	2.0164	0.2868	7.032	7.511	**<0.0001**
**aEUA**						
Parameter	*n*	Estimate	Standard error	z-value	OR	*p*
Size (<2.5 cm vs. ≥2.5 cm)	296	−0.5839	0.3771	−1.548	0.56	0.122
BMI (<30 kg/m^2^ vs. ≥30 kg/m^2^)	392	0.2648	0.4078	0.649	1.3	0.516
Parametrial status (MRI, negative vs. positive)	400	2.0164	0.2868	7.032	7.511	**<0.0001**
**Multivariable Regression Modeling**						
**MRI**						
Parameter	*n*	Estimate	Standard error	z-value	OR	*p*
Size (<2.5 cm vs. ≥2.5 cm)	296	−1.4885	0.4188	−3.554	0.23	**0.00038**
BMI (<30 kg/m^2^ vs. ≥30 kg/m^2^)		0.2475	0.3995	0.619	1.28	0.53565
Parametrial status (clinical exam, negative vs. positive)		1.7899	0.3287	5.445	5.99	**<0.0001**
**aEUA**						
Parameter	*n*	Estimate	Standard error	z-value	OR	*p*
Size (<2.5 cm vs. ≥2.5 cm)	296	−0.07903	0.41051	−0.193	0.92	0.847
BMI (<30 kg/m^2^ vs. ≥30 kg/m^2^)		0.51993	0.48890	1063	1.68	0.288
Parametrial status (MRI, negative vs. positive)		1.7899	0.3287	5.445	5.99	**<0.0001**

BMI: body mass index. aEUA: Examination under anesthesia with display of MR images. CI: confidence interval. MRI: magnetic resonance imaging. NPV: negative predictive value. OR: odds ratio. PPV: positive predictive value. Bold: highlight significant values.

## Data Availability

The data presented in this study are available on request from the corresponding author. The data are not publicly available because it contains sensitive patient information.

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
