# Peer review of "The Importance of Clinical Examination under General Anesthesia: Improving Parametrial Assessment in Cervical Cancer Patients"

_cancers, 2021, doi:10.3390/cancers13122961_

Round 1

Reviewer 1 Report

The authors investigated the diagnostic value of clinical examination under general anesthesia (EUA) and magnetic resonance imaging (MRI) in determining parametrial tumor spread in patients with cervical cancer.

I have the following concerns/comments:

SIMPLE SUMMARY:

  1. There is a typo - “comparted” -> compared

ABSTRACT.

  1. The conclusion should meet the aim of the study. Therefore, the aEUA’s added value should be concluded related to the parametrial tumour involvement.

TITLE

  1. The title should be informative and attractive, for instance providing a statement or a question (Ex: Can clinical examination under anesthesia improve the detection of parametrial involvement by cervical cancer?)

MATERIAL AND METHODS

  1. Patient selection. What was the timeline between the MRI and EUA? – As this could affect the results.

RESULTS

  1. There is an inconsistency regarding the study period. In the results section: “During the study period from 10/1999 – 05/2017”, while in the Materials and Methods section it is written: “between 10/2000 and 07/2017”. This should be corrected.
  2. In theory, the clinical assessment does not include intraoperative assessment, and it is based on the gynecology examination. The authors should be consistent and clear and should use the same term for their “clinical assessment” which is in fact an intraoperative + imaging assessment (if I understood correctly). The paragraph “In summary, clinical assessment of the parametrium showed a statistically significantly higher sensitivity, specificity, PPV, and NPV regarding tumor involvement as compared to MRI alone » should be rephrased.
  3. “We found that sensitivity and PPV of MRI increased from 39.5% and 35.7%, respectively, in cases without clinical evidence of parametrial involvement (n=243, 60.8%) to 77.2% and 85.2% in tumors staged clinically FIGO IIB or higher (n=157, 39.2%, p<0.0001 for both, sensitivity and PPV) ».-> The paragraph is not clear enough. What does it mean cases without clinical evidence, but clinically staged IIB? FIGO staging is a surgical staging, and the authors mention in their study “clinical examination” as an intraoperative assessment.
  4. The same comment for “clinically suspected parametrial infiltration”.
  5. Expressions like “best result” should be avoided. Also, the author should keep the term “Finally” for the conclusion and not when presenting the results. Please rephrase.
  6. There is a typo:” In 315 (78.8%) congruent cases, parametrial involvement was recognized with a sensitivity of 81.7% and specificity was 91.3% ». Please correct.

DISCUSSION

  1. “This current study was conducted to investigate the accuracy of parametrial assessment in patients with cervical cancer using MRI and clinical examination” -> please add intraoperative clinical examination.
  2. “In one former meta-analysis, pooled sensitivity of PMI assessment in the clinical examination was 40%. “ – Reference should be added.

Reviewer 2 Report

This is overall an interesting study presenting a large study population from a single university hospital treating cervical cancer. The fact that all patients in the study are surgically treated with histological assessment of parametrial invasion (PMI) makes it quite unique and allows access to a histological gold-standard for PMI that is normally not available in patients with advanced FIGO stage. The manuscript is well written and the results are overall nicely structured, presented and discussed. Also the main conclusion that aEUA yields more accurate staging for PMI than MRI alone seems justified.

However, there are some issues that need to be addressed:

Since the clinical examination under anesthesia (EUA) was performed by gynecologists having access to the MRI results, this study is not able to present diagnostic performance of clinical EUA alone for comparison with that of MRI alone for diagnosing PMI. The diagnostic performance metrics for aEUA is inherently influenced by the MRI results, and is thus expected to yield better performance metrics than that of EUA alone. This needs to be more clearly stated and discussed as a possible limitation and the discussion on e.g. high sensitivity of 77% of clinical examination in this study needs to be modified.

The findings regarding impact of tumor size on the derived diagnostic performance metrics for diagnosing PMI is interesting. However, information on the prevalence of PMI for the patient categories with tumors < and >2.5 cm should be reported, since this may impact the interpretation of the results (somewhat touched upon in the discussion). 

Minor issues:

Methods page 2 (last line) and 3 (first line): “The radiology reports were not available during clinical assessment”. How is this possible when the images were displayed intraoperatively? Did the clinicians interpret the MRI findings themselves without support from a radiology report? Please clarify.

Last sentence in first paragraph page 3: “In addition, we investigated whether the sensitivity and specificity of parametrial assessment….” This sentence is too long – please rephrase to improve clarity.

Tables:

Percentages should be given without decimals to improve readability. This applies to all tables and text in result.

Table 1: List paraaortic lymph nodes with pM0 before pM1 (similar to that of pelvic lymph nodes).  

Table 2-5: Please add more informative titles: e.g. for Table 2 “Diagnostic performance metrics of MRI and aEUA for diagnosing of PMI at histology”. I suggest adding metrics for accuracy in Tables 2-4 to include this central information in the Tables.

Reviewer 3 Report

General comments:

This is an interesting and important study investigating the role of clinical pelvic examination under general anesthesia (EUA) in predicting parametrial involvement. The study results is relevant in the wake of the new FIGO 2018 guidelines that are now including MRI assessments in staging. Accurate pre-operative assessment of parametrial involvement is important for treatment decisions.

In general, the manuscript is well written although some typos exist. The introduction gives a good background. The methods are clearly presented, but it lacks explanations of why the cut offs for tumor diameter and BMI are chosen (See Specific Comments). The results are a bit hard to follow. In the text, you are jumping from table to table. Please structure the results part better and present one table at the time if possible. The discussion is well written and mentions some important limitations in the study. The conclusion part is in my opinion, not quite accurate. If you test for MRI scanner, algorithm and time period and find that these claims are still valid, I will agree with the current text (See more details in Specific comment 4).

Specific comments:

  1. I think it would be easier to read the tables if you had a row at the bottom showing the accuracy for each modality.
  2. Please explain why you chose 2.5 cm maximum diameter at cut off for small versus large tumors. I think you should stick to 2 or 4 cm as cut offs as these are already implemented in the new FIGO guidelines. If you want to have another cut off, you need to explain why you chose that one, or do a statistical test to uncover at what tumor size the accuracy starts to decline.
  3. Please explain why you chose 30 (as opposed to e.g. 25) as cut-off for BMI.
  4. I share your concern regarding the generalizability of the results as the inclusion of patients involved almost two decades and MRI protocols have changed and probably improved along the way. The earliest MRI scans in this cohort could thus negatively influence the accuracy and overestimate the difference between MRI alone and MRI assisted clinical examination. I would suggest that you test whether MRI scanner and algorithm is influencing the accuracy, and also whether year of diagnosis is playing a part.

Reviewer 4 Report

The diagnostic value of clinical examination under general anesthesia (UAE) and magnetic resonance imaging (MRI) in determining parametrial tumor spread has been presented in a very large series of cervical cancer.

The paper is well written and supported by good quality statistics. However, I have some perplexity about research design.

In the introduction, the authors state that MRI is known to be better than UAE in early cervical cancer (ECC), while the role in Locally advanced cervical cancer (LACC) is not clarified. But then they present a series of which > 40% are IB1 stage. Perhaps if the need to be clarified is that in the LACC, the ECCs should be removed from the analysis. Please consider rephrasing the Introduction, eventually referring to different papers.

In my opinion, the fact that gynecologists have MR images in front of them while performing UAE is a very big bias (not only a limit) of the study. I find it difficult to assert the superiority of one method over another when the first is supported by the data presented in the second.  If the radiologists had been able to report their MR having the UAE report in front of them, the results might have been different. Usually, if I know the answers I can improve my performance, it is not likely that I can do worse.

It is not clear to me the rationale for which the authors question whether MR or UAE is better in locally advanced cervical cancers. Furthermore, since the standard treatment is exclusive radiochemotherapy, not surgery, these results are not very applicable in the clinical practice of treating locally advanced cervical cancers. In the discussion, the clinical settings in which the results of this analysis can be useful could be afforded.

Round 2

Reviewer 1 Report

I am supportive of publishing this study in view of the importance of the problem but still remain a little concerned about the quality of the examinations included, considering the study period: 2000-2017. Otherwise, the authors performed the necessary changes. 

Author Response

We would like to thank the reviewer for again going over our manuscript.

We agree with the reviewer, that the long study period entails risks for bias, however, we explicitely state this problem in our discussion so every reader will be aware of this issue. Maybe, this point will also be raised in future correspondence with readers. 

At the same time, limiting our study period to a shorter time frame would significantly reduce the number of patients, leading to less statistical power. 

As another prospective study addressing this issue is underway, we don't think that further methodic changes are indicated and appreciate the reviewer's support for the publication of our study.

Reviewer 4 Report

The authors have added several comments that improve the objectivity of their findings, therefore I believe that the paper in this form can be accepted for publication. 

Author Response

We would like to thank the reviewer for re-examining our manuscript. As no specific remarks have been made, we have just performed another spell check and identified a few minor typos. We have corrected these and highlighted the changes in the manuscript.